# Beyond the Clinic: Maximum Free-Living Stepping as a Potential Measure of Physical Performance

**DOI:** 10.3390/s23146555

**Published:** 2023-07-20

**Authors:** Craig Speirs, Mark D. Dunlop, Marc Roper, Malcolm Granat

**Affiliations:** 1PAL Technologies Ltd., Glasgow G4 0TQ, UK; 2Department of Computer and Information Sciences, University of Strathclyde, Glasgow G1 1XH, UK; mark.dunlop@strath.ac.uk (M.D.D.); marc.roper@strath.ac.uk (M.R.); 3School of Health and Society, University of Salford, Salford M6 6PU, UK

**Keywords:** physical activity, accelerometer, activPAL, iData, stepping

## Abstract

Measures of physical performance captured within a clinical setting are commonly used as a surrogate for underlying health or disease risk within an individual. By measuring physical behaviour within a free-living setting, we may be able to better quantify physical performance. In our study, we outline an approach to measure maximum free-living step count using a body-worn sensor as an indicator of physical performance. We then use this approach to characterise the maximum step count over a range of window durations within a population of older adults to identify a preferred duration over which to measure the maximum step count. We found that while almost all individuals (97%) undertook at least one instance of continuous stepping longer than two minutes, a sizeable minority of individuals (31%) had no periods of continuous stepping longer than six minutes. We suggest that the maximum step count measured over a six-minute period may be too sensitive to the adults’ lack of opportunity to undertake prolonged periods of stepping, and a two-minute window could provide a more representative measure of physical performance.

## 1. Introduction

There are clinical measures of physical performance, including the six-minute walk test and timed up and go test, that have a strong evidence base demonstrating their relationship with health outcomes, both within the general population and in clinical subpopulations. However, due to the need for appropriately trained staff to manage these tests and the distance that participants may be required to travel to attend test sites, these clinical measures can be difficult to administer across larger clinical populations. The development of free-living analogues of these measures may be more resistant to participant bias and could be suitable for the development of novel digital biomarkers.

Wearable devices have been shown to accurately measure a range of posture and gait characteristics, including step count [1], cadence [1] and sedentary and upright time [2]. They are increasingly used in healthcare to capture physical behaviour in a free-living setting over a prolonged period of time, providing information that would be difficult to quantify within a clinical setting. This information may enable the development of outcomes based on free-living behaviour, providing clinical insights that are not apparent from existing clinical measures.

The six-minute walk test is an exercise test that measures aerobic capacity by assessing the maximum distance an individual can walk during a six-minute period. The protocol for the test is well defined [3] and has been found to have predictive value in a number of conditions, including cancer [4], cardiopulmonary disease [5] and heart failure [6]. The two-minute walk test is a shorter variant of the test, measures the maximum distance an individual can walk during a two-minute period and is used in clinical populations who are unable to tolerate the prolonged periods of walking required for the six-minute walk test [7,8]. An analogue of the six-minute walk test using free-living data may provide a more representative measure of physical performance by removing the measurement limitation seen in the six-minute walk test [3], while also reducing the financial and administrative cost for both the organiser and test participant. As thigh-worn accelerometers can identify individual strides within a period of stepping, it should be possible to quantify the maximum number of steps an individual takes within a defined time period as a free-living analogue of the six-minute walk test. 

This study aims to explore the potential of a free-living equivalent of the six-minute walk test, the maximum n-minute step count, using accelerometer data obtained from a population of older adults.

## 2. Materials and Methods

### 2.1. Design and Participants

The Interactive Diet and Activity Tracking in AARP (iDATA) study, funded by the National Institute of Health (NIH), recruited individuals between the ages of 50 and 74 who were members of AARP, a non-profit organization that supports older adults in the USA and who were living in Pennsylvania [9]. Over a 12-month period, participants undertook three clinical visits and carried out a variety of home-based activities, including capturing physical activity data [10]. In total, 1082 participants took part in the study, and their characteristics are presented in Table 1.

### 2.2. Physical Activity Measurement

As part of the study, participants wore a thigh-mounted triaxial accelerometer (activPAL3 micro; PAL Technologies Ltd., Glasgow, UK), to capture two periods, separated by six months, of seven days of continuous free-living physical activity data. The raw accelerometer data was then processed by the National Institute of Health using PALanalysis (version 8.4.9.19) to extract the data in an event-based format. In this format, each continuous period of a specific type of activity, for example, sitting, standing, lying down and taking a stride, is considered a single event. Each detected **stride event** comprises two steps.

Subsequently, all adjacent stride events were combined into a single event, termed a “**stepping bout**”. This allowed periods of stepping to be characterised by duration, step count and cadence (defined as mean steps per minute). To characterise contiguous blocks of events that could contribute towards the definition of a period of free-living stepping, **upright containers** were then defined by combining adjacent standing and stepping events that were not interrupted by a sedentary (sitting or lying down) event.

### 2.3. Data Cleaning and Statistical Analysis

In this project, initial cleaning of the activity data was performed using R, a programming language used in statistical and data analysis [11]. To remove days that did not have complete physical activity data, a day was only considered valid and included in the analysis if it contained more than 10 h of classified activity data and more than 500 steps. This is consistent with previous studies investigating physical activity data from the iData study [12,13]. Skew, kurtosis and intraclass correlation coefficients were calculated using the R library psych (R psych package, version 2.2.5) [14].

In order to assess availability of stepping bouts of different durations of the continuous free-living walking data, we calculated the maximum n-minute step count for each valid day by passing a search window across all upright containers to calculate all possible n-minute step counts (Figure 1, example 1 and 2). When a search window encompasses part of a stepping bout (Figure 1, example 3), only stepping within the window contributed to the step count. If a break in being upright is encountered within the search window (Figure 1, example 4), only stepping occurring prior to this break was included in the step count.

The maximum step count was then selected from all the calculated n-minute step counts. If more than one search window period contained the maximum step count, the window which accumulated the maximum step count in the shortest time was chosen. In cases in which multiple search windows accumulated stepping in the same duration, the first search window was selected. 

We used this approach to characterise the maximum number of steps for different window lengths, ranging from 30 s to 10 min. Peak step counts were also expressed as the step accumulation rate (peak step count divided by search window duration) to facilitate comparison across different search window durations.

For each individual, we also calculated the maximum step count across all window durations within the first seven days of each observation period, the second occurring six months after the initial observation. In cases in which an individual participated in both periods of observation, we only included the first observation which had seven days of valid data in this analysis. For each day classified as valid during the observation period, for each of the n-minute windows considered, we calculated the number of distinct stepping bouts whose duration was equal to or exceeded the search window duration.

In order to incorporate a tolerance for periods of upright activity that contained multiple, but interrupted, stepping bouts, for each individual we also calculated the duration of stepping within the period they completed their maximum n-minute step count. For each search window duration, we then calculated the **proportion** of individuals whose stepping time was longer than a given proportion of the window duration (100%, 95%, 90% and 80%, respectively).

For each n-minute period, we calculated the distribution of maximum n-minute step count for the study population. The distribution was classified as normal if skewness was below two and kurtosis below seven [15]. In order to test for seasonal effects on maximum step count, maximum n-minute step counts were stratified by month, and a one-way ANOVA was performed to compare the effect of observation period on maximum step count.

## 3. Results

Out of the 1082 individuals who participated in the study, 914 (84%) had at least one observation period with 3 valid days of activity data, 892 (82%) had at least one observation with 5 valid days and 814 (75%) had at least one observation containing 7 days of valid data encompassing the entire observation period. The majority of the participants identified as white, non-Hispanic (91%) and were overweight or obese (73%). There was no significance difference in age or body mass index (BMI) between males and females (*p* = 0.064).

### 3.1. Peak n-Minute Step Count

Figure 2 shows the proportion of participants who had one or more n-minute period of activity in which the time spent stepping comprised more than a specified proportion of the period. Almost all the participants had at least one stepping bout that was at least two minutes (97%); however, as the duration of the stepping bout increases, a decreasing proportion of the population had at least one instance of continuous stepping of the desired duration (from 89% for three minutes of continuous stepping to 65% for six minutes of continuous stepping, with 55% of individuals achieving ten minutes of continuous stepping). As the proportion of required stepping within the window decreases, the proportion of individuals with a single occurrence of qualifying stepping increases across all the search windows. However, even when allowing for increasing durations of non-stepping behaviour, a sizeable proportion of individuals did not undertake any periods of prolonged stepping. For example, 10% of the participants did not have any six-minute periods of upright activity containing continuous stepping, even when allowing for the presence of up to 72 s of quiet standing.

The distribution of the stepping bout durations within the population is shown in Figure 3. The peak step counts were normally distributed across all of the search window durations (skew: −0.2 to 0.7, kurtosis: −0.7 to 0.1). For stepping bout durations of five minutes or longer, participants tended to undertake less than one stepping bout of that duration each day. Among participants who undertook continuous stepping longer than the period of interest, a proportion only undertook a single bout of continuous stepping exceeding this duration (9% for five minutes of continuous stepping to 13% for ten minutes of continuous stepping).

### 3.2. Maximum Six-Minute Step Count

Given the established six-minute walk test and the related two-minute walk test durations, we looked in more detail at the availability of walking bouts for these duration in the free-living data. The maximum six-minute step count was normally distributed (skew = −0.2, kurtosis = −0.2, Figure 4). The maximum six-minute step count was 637 + 142 (range: 142–992). There was no significant difference in the maximum six-minute step count between males (641 + 156) and females (633 + 151) (*p* = 0.47). No significant difference in the maximum six-minute step count was observed across different months of the year (F(11,802) = 1.44, *p* = 0.15).

### 3.3. Maximum Two-Minute Step Count

The maximum two-minute step count was normally distributed (skew = 0.4, kurtosis = 0.7, Figure 5). The maximum two-minute step count was 239 + 35 (range, 106–340). There was no significant difference in the maximum two-minute step count between males (238 + 38) and females (240 + 33) (*p* = 0.37). No significant difference in the maximum two-minute step count was observed across different months of the year (F(11,802) = 1.63, *p* = 0.08).

The maximum two-minute step count was strongly correlated with the maximum six-minute step count (r(812) = 0.771, *p* < 0.01) (Figure 6) within our population. For individuals with a lower maximum six-minute step count, there was a visible increase in the variation in the maximum two-minute step count. Across individuals within our population, the maximum two minute step accumulation rate (M = 120, SD = 17) was significantly higher than the maximum six minute step accumulation rate (M = 106, SD = 24) (t(1454) = 12.4, *p* < 0.05).

## 4. Discussion

This study aimed to explore the potential of a free-living equivalent of the lab-based six-minute walk test using physical activity data derived from a thigh-worn accelerometer. By using a window-based search strategy, we characterized the maximum step count in our population of older adults across a range of stepping durations.

Given the existing evidence for the effectiveness of the six-minute walk test as a biomarker in a range of medical conditions, a six-minute window would appear to be a suitable duration to capture a free-living maximum step count. In studies that examined similar populations’ performance in the six-minute walk test, almost all individuals were able to walk unassisted for the entirety of the six-minute period [16,17], with an early termination of the test indicating poor aerobic capacity. However, within our population, nearly one in five individuals with seven days of physical activity data had no instances in which they walked for six minutes, with less than half of our population undertaking on average more than one bout of six minutes of continuous stepping each day. Within a free-living environment, this may be due to functional limitations or the absence of opportunities to undertake the required duration of stepping, regardless of physical capacity. We suggest that six minutes may be too long for a free-living maximum step count, as it is likely to lead to the misclassification of healthy individuals who are functionally capable of walking continuously for six minutes but have limited opportunities to undertake prolonged stepping during the observation period.

We suggest a suitable duration for a peak step count outcome should maximise the proportion of individuals who undertake multiple periods of stepping of the required duration. As there are a range of free-living activities associated with continuous stepping undertaken at sub-maximal cadences, including shopping-associated walking and dog walking, a duration that captures multiple distinct periods of stepping should increase the robustness of the maximum step count as a measure of functional ability, assuming that a sufficient range of step counts that are reflective of differences in ability, by increasing the likelihood that at least one of the periods of stepping provides a good indication of aerobic capacity.

Our analysis of the proportion of individuals undertaking continuous stepping across different durations suggests that two minutes may be an appropriate duration for a measure of free-living maximum step count, as it both covers a wider range of participants and strongly correlates with the six-minute data in the study for those who achieved six minutes. This minimises the number of individuals who do not undertake any continuous stepping longer than the search window, while allowing for a sufficient period of stepping activity to provide a range of maximum step counts for the effective stratification of individuals to identify subgroups of interest.

Within our study population, almost all individuals undertook at least one bout of stepping that was longer than two minutes, with individuals on average undertaking more than three distinct daily periods of stepping longer than two minutes each day. This should provide a sufficient number of observations to calculate a maximum step count that is likely to be a robust measure of aerobic capacity. An additional advantage of using a two-minute threshold is the absence of sex-specific differences in the maximum step count that can be seen in the six-minute walk test [17]. This should make the maximum step count more robust as sex is less likely to be a confounding factor when investigating the relationship between the maximum step count and diseases or biological processes of interest.

While the strong evidence demonstrating the predictive and diagnostic power of the six-minute walk test guided our initial selection of six minutes as the duration for our free-living maximum step count, studies have found that the distance achieved in the two-minute walk test tends to be correlated with distance achieved in the six-minute walk test [18,19]. This suggests the two-minute window we suggest for calculating the maximum step count should also have face validity based on existing evidence about the effectiveness of the two-minute walk test.

### 4.1. Strengths

This study’s main strengths include the size of the sample, which allowed us to quantify the characteristics of the maximum n-minute step count across different window durations for a diverse range of individuals. Another strength of the study is that each participant has two distinct periods in which their physical activity is captured. This increases the number of individuals who have seven days of valid activity data, allowing us to maximise the number included in our analysis.

### 4.2. Limitations

A limitation in our study is that the study sample comprised a non-representative self-selecting sample of older adults residing in a geographically limited location. Our population was also older and more racially homogenous than the general population of the USA. It is likely that our population has a lower exercise capacity and general fitness compared to the general population, with fewer opportunities for vocational-based physical activity given that a significant proportion of the participants are likely to be retired. We aim to test the validity of the maximum two-minute step count in other populations with different demographics.

A further limitation in our study arises from the inclusion of periods of running when identify the maximum step count, whereas in the lab-based six-minute walk test, participants are explicitly instructed not to jog or run [3]. In our population, a sizeable number of individuals have a maximum step count above 300 steps (Figure 5), whose cadence (>150 steps per minute) is suggestive of running during the identified period of activity. The absence of running during the observation period may not be an indication that an individual is unable to run for a prolonged period. We must therefore account for differences in the maximum step count caused by periods of running, as these are likely to be driven by the opportunity and requirement to run, as opposed to underlying functional limitations preventing an individual from running.

While our development of a free-living maximum two-minute step count is based on its similarity to a validated measure of aerobic fitness, the six-minute walk test, there is currently no evidence demonstrating that the maximum free-living two-minute step count is correlated with the distance completed in the six-minute walk test. If there is no relationship between these measures, we would be unable to leverage existing evidence about the six-minute walk test to identify conditions in which the maximum two-minute step count could be used. A follow-up study measuring both indicators would allow us to test if the maximum two-minute step count is correlated with the results of the six-minute walk test. A validation study using data obtained from an instrumented walkway paired with free-living physical activity data obtained using a body-worn activity monitor would allow us to test the accuracy of the monitor-measured step count in our free-living maximum step count.

## 5. Conclusions

This study tested the effectiveness of a range of durations for a free-living maximum step count measure. Our analysis suggested a two-minute threshold provides a wide range of step counts that would allow for the stratification of individuals within a population. This threshold also appears to minimise the confounding impact of healthy individuals having limited opportunities to undertake prolonged periods of stepping, which is more apparent when longer thresholds are used. Future work will characterise the maximum step count over a two-minute period in other populations, testing its repeatability and effectiveness as a digital biomarker.

## Figures and Tables

**Figure 1 sensors-23-06555-f001:**
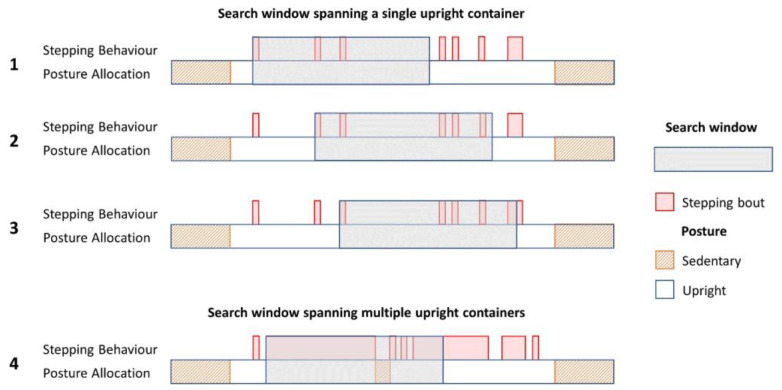
Examples of calculating total step count using a sliding window approach. The lower bar shows the posture of the individual, and the upper bar gives distribution of stepping activity within the upright container. Bar width is used to denote the duration of each stepping bout. The period of activity considered within the current search window is represented by the semi-transparent box.

**Figure 2 sensors-23-06555-f002:**
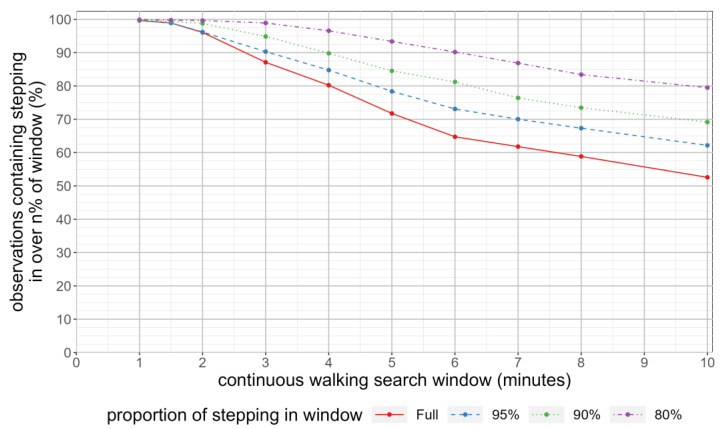
Proportion of individuals with at least one instance with a window of activity composed of differing proportions of stepping.

**Figure 3 sensors-23-06555-f003:**
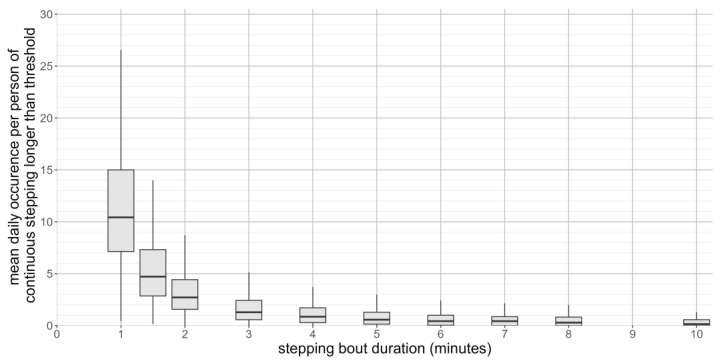
Mean daily occurrence of continuous stepping bouts of different duration during observation period (seven-day observation period).

**Figure 4 sensors-23-06555-f004:**
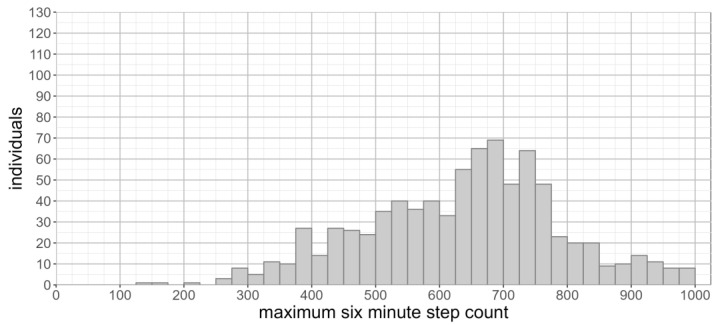
Distribution of maximum six-minute step count (seven-day observation period).

**Figure 5 sensors-23-06555-f005:**
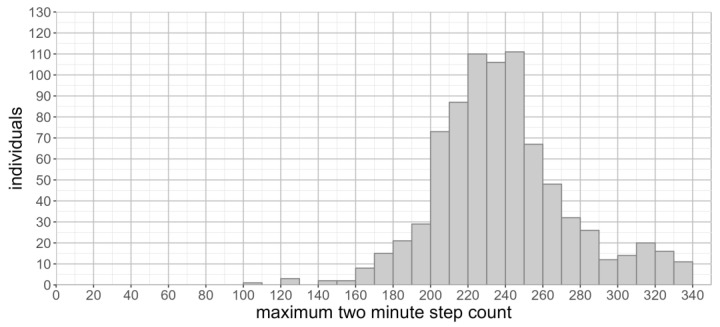
Distribution of maximum two-minute step count (seven-day observation period).

**Figure 6 sensors-23-06555-f006:**
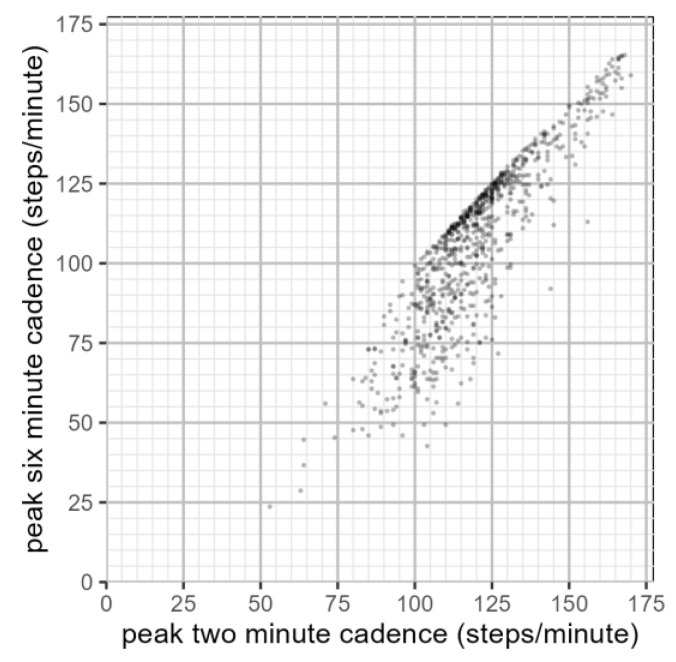
Per-individual relationship between peak two-minute step accumulation and peak six-minute step accumulation (seven-day observation period).

**Table 1 sensors-23-06555-t001:** Characteristics of participants, iData cohort.

Variable		Mean (SD)		
		Male	Female	Total
N		541	541	1082
Age, years		64.0 (5.7)	62.2 (6.1)	63.1 (6.0)
Race	White, non-Hispanic	514	476	990
	African American	21	58	79
	Hispanic	3	2	5
	Asian	3	4	7
	Unspecified	0	1	1
BMI		28.4 (4.3)	27.9 (5.0)	28.2 (4.7)

## Data Availability

Restrictions apply to the availability of these data. Data were obtained from the National Cancer Institute and are available at https://cdas.cancer.gov/idata/ with the permission of the National Cancer Institute.

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
