# Peer review of "Beyond the Clinic: Maximum Free-Living Stepping as a Potential Measure of Physical Performance"

_sensors, 2023, doi:10.3390/s23146555_

Round 1

Reviewer 1 Report

Please see attached comments.  The data seems to support differences in maximum step count between 6-minute vs 2-minute walk test after normalization and that one may not be a proxy for the other. It does contradict the key finding that the 2-minute walk test in the home environment is a good proxy for the 6-minute. The main finding seems to be that there are more 2-minute windows of walking than there are 6-miute windows, which is not so surprising... or clinically relevant.

Author Response

Please see the attached word document with detailed repsonses to your comments.

Reviewer 2 Report

Dear Authors,

This study aims to explore the potential of a free-living equivalent of the six-minute walk test, the maximum n-minute step count, using accelerometer data obtained from a population of older adults.

Currently, the problem of free-living behavior evaluation is extensively investigated and covered in many similar publications. The reference list should be revised. Too many old references are presented. In addition, the novelty of the study should be presented clearer.

The major comment: The comparison between activity data obtained in various time-periods was not performed or clearly presented. So, the conclusions, not fully supported by the results. The title of the manuscript does not match the results and needs to be edited.

In order better understanding, the list of primarily registered parameters and following calculated indices would be useful. Please, check that all registered parameters were analyzed and presented in the results section.

The table 2 should be explained. Since the study was aimed at comparing two-minute and six-minute walking activity, why were the data presented in the range of 30 seconds - 10 minutes.

In my opinion, there is no need to repeat the sentence about subjects. Line 128-129 “The study participants' characteristics are presented in table 1” can be deleted.

If no differences were found between males in females data could be presented as a total group.

Line 162-163: no difference was fond in maximum six-minute step count between males in females. The sentence “Maximum six-minute step count was higher in males (641 + 156) compared to females (633 + 151), but this difference was not significant (p = 0.47)” needs to be edited.

Line 171-172: no difference was fond in Maximum two-minute step count between males in females. The sentence “Maximum two-minute step count was lower in males (238 + 38) compared to females (240 + 33), but this difference was not significant (p = 0.37)” needs to be edited.

Keep the unified terminology (step count or stepping count) across the manuscript. 

My overall comment: The manuscript in its present form is not ready for publishing. There are flaws in the methodology and presentation of results in the article that need to be corrected

Author Response

(The authors gave the same response as above.)

Round 2

Reviewer 1 Report

Many thanks to the authors for addressing the items listed in the prior review.  All concerns have been addressed in the revision and this manuscript is ready for publication.

Reviewer 2 Report

The manuscript was sufficiently improved.